# Graphene nanomechanical vibrations measured with a phase-coherent software-defined radio
Ce Zhang[1,2], YuBin Zhang[1,2], Chen Yang[1,2], Heng Lu[1,2], FengNan Chen[1,2], Ying Yan[1,2] & Joel Moser [1,2] ✉

Software-defined radios (SDRs) are radio frequency transceivers designed to facilitate digital signal processing through the use of vast libraries of open-source software. Here, we assemble a simple data acquisition system whose architecture, based on SDR, allows us to develop a comprehensive suite of tools to study the vibrations of a few-layer graphene nanomechanical resonator. Namely, we measure the cross-spectrum of vibrations in the frequency domain, we measure their energy decay rate in the time domain, we perform vector measurements of their in-phase and quadrature components, and we control their phase using a time-dependent strain field –all with a single measurement platform. Our approach allows us to tailor our experiments at will and gives us control over every stage of data processing. Overall, our versatile system enables measuring a wide range of nanomechanical properties of graphene by customizing the signal acquisition and replacing some analog electrical circuits, such as filters, mixers, and demodulators, by blocks of code.

Graphene nanomechanical resonators are the thinnest vibrating membranes imaginable[1–4]. They play an important role in the study of the flexural vibrations of two-dimensional (2-D) systems[5]. Because of their small thickness and their small mass, they constitute a nearly ideal testbed for studying a wealth of nanomechanical phenomena[6]. They also lend themselves to interesting sensing applications[7], where the lightness of the membranes enables the detection of weak ultrasound[8,9] and tiny optical power[10]. To address these fundamental and applied topics, researchers have overcome the challenge of detecting vanishingly small vibrations by devising a variety of measurement techniques. These range from optical absorbance measurements[1] to electrical transport measurements[2,11] to Fizeau interferometry measurements[4] to capacitive measurements[12–14]. These techniques involve the use of various data acquisition instruments, each carefully designed for a particular purpose. For example, electrical transport measurements require a lock-in amplifier to measure the in-phase and quadrature components of narrow band signals, while optical and capacitive measurements may require an oscilloscope, or a digitizer, or a spectrum analyzer to acquire signals in the time or in the frequency domain. Recently, researchers in the field of magnetic resonance force microscopy (MRFM), where a micromechanical cantilever is used as a sensitive force probe to detect electron and nuclear spins[15], have pursued the idea of grouping these various measurement functions within a single system[16]. While every

nanomechanical experiment is rather complex, MRFM ones are particularly demanding: not only does one need to measure the spectrum of the cantilever's weak vibrations, but one also has to keep the amplitude of the vibrations constant and adjust their phase dynamically in a feedback control loop[17,18]. The single system devised by MRFM researchers leverages the versatility, parallel processing capabilities and high computation speed of a digital signal processing hardware platform to implement functions as diverse as those of a lock-in amplifier, a spectrum analyzer, and a controler. This system makes it easier to operate the experimental setup and also offers technical advantages. Inspired by this approach, we have assembled a simple and versatile data acquisition system that allows us to study various aspects of the dynamics of a few-layer graphene (FLG) nanomechanical resonator. The versatility of our data acquisition system stems from its software-defined radio (SDR) architecture. Below, we provide a brief overview of SDRs in a context that is directly relevant to our work.

SDRs aim at manipulating and controlling the radio spectrum while performing as much signal processing as possible in software[19]. These devices are essential components in cognitive radio systems, allowing unused frequency bands to be detected and dynamically allocated to facilitate communication within a crowded radio spectrum[20]. SDRs are also employed in a range of scientific applications. These include radio astronomy, where custom-built SDRs are used as multichannel receivers to

[1]School of Optoelectronic Science and Engineering & Collaborative Innovation Center of Suzhou Nano Science and Technology, Soochow University, Suzhou, China. [2]Key Lab of Advanced Optical Manufacturing Technologies of Jiangsu Province & Key Lab of Modern Optical Technologies of Education Ministry of China, Soochow University, Suzhou, China. ✉e-mail: j.moser@suda.edu.cn

process data collected by arrays of radio telescopes[21], and smaller scale experiments about oscillator metrology[22], optoelectronics[23], magnetic resonance spectroscopy[24], and optical spectroscopy[25], where commercially available SDRs are used. The architecture of SDRs reflects the need for a radio system that is compact, immune to environmental change (such as filter response drifts caused by temperature changes), and whose modulation and demodulation stages are highly reconfigurable. The front-end module of SDR receivers replaces analog superheterodyne circuits with real-time digital down-conversion. Namely, analog local oscillators are replaced by digital sinusoids either generated by numerically controlled oscillators or computed by a field-programmable gate array (FPGA); analog mixers are replaced by computed multiplications of the digitized input signal and the digital sinusoids; and analog low-pass filters at the mixers' intermediary frequency are replaced by digital filters and decimators. Importantly, digital down-conversion of an input signal using two sinusoids that are 90° out of phase results in two output streams of data representing the down-converted in-phase $I$ and quadrature $Q$ components of the input signal. In MRFM experiments based on SDRs, for example, one of the sinusoids is in phase with a reference signal at the resonant frequency of the cantilever, making it possible to extract the $I$ and $Q$ components of the cantilever vibrations. From the above description, it appears that SDRs share some basic features with better-known wideband digitizers. However, the originality of SDRs is their reconfigurability, which is enabled by libraries of open-source software. It is the free access to and the unrestricted use of a vast collection of drivers and data processing applications that allow users to build their own software-defined instruments, sometimes rivaling in performance the dedicated hardware instruments they set out to emulate[16,26].

In this work, we measure the flexural vibrations of a nanomechanical resonator made of suspended FLG using a phase-coherent data acquisition system based on SDR. The phase coherence of the acquisition means that digitized waveforms of coherent signals acquired sequentially are all in phase with each other, and that the phase difference between two signals acquired simultaneously is nearly constant (it is either $\simeq 0$ or $\simeq 90°$). Implementing this important feature was not possible in our earlier work[27], where we employed a single-channel, narrow-band SDR dongle as a radio frequency power meter to measure large amplitude vibrations in a different type of FLG resonator. Here, the phase coherence and the versatility of SDRs allow us to perform 4 different types of nanomechanical measurements using a single platform. These are: (i) the measurement of weakly driven vibrations, including their cross-spectrum in the frequency domain and their vibrational amplitude ringdown in the time domain; (ii) the measurement of the cross-spectrum of vibrations actuated by a force noise; (iii) the vector measurement of $I$ and $Q$; and (iv) the measurement in the time domain of vibrational phase modulation induced by strain modulation. To illustrate the advantages of our SDR-based approach, we compare vector measurements of vibrations performed with analog heterodyning on the one hand and with digital down-conversion on the other hand. In addition to its flexibility and its versatility, our simple data acquisition system allows us to emulate ordinary instruments that used to be a common occurrence in laboratories but are now difficult to find on the market. This is especially the case for cross-spectrum measuring instruments, whose purpose is to extract small signals out of a large noise background[28]. Our system may also be useful where access to radio frequency equipment is restricted. We have also found that our approach has its own educational merit, as it appeals to highschool and university students and facilitates the teaching of radio frequency physics and engineering.

## Results and discussion
### Experimental setup
Our data acquisition system is based on the SDR receiver SDRlab 122-16 by Red Pitaya (Fig. 1a)[29–31]. (For some specific applications, we replace this receiver with a digital oscilloscope.) SDRlab comes with two 16 bit analog-

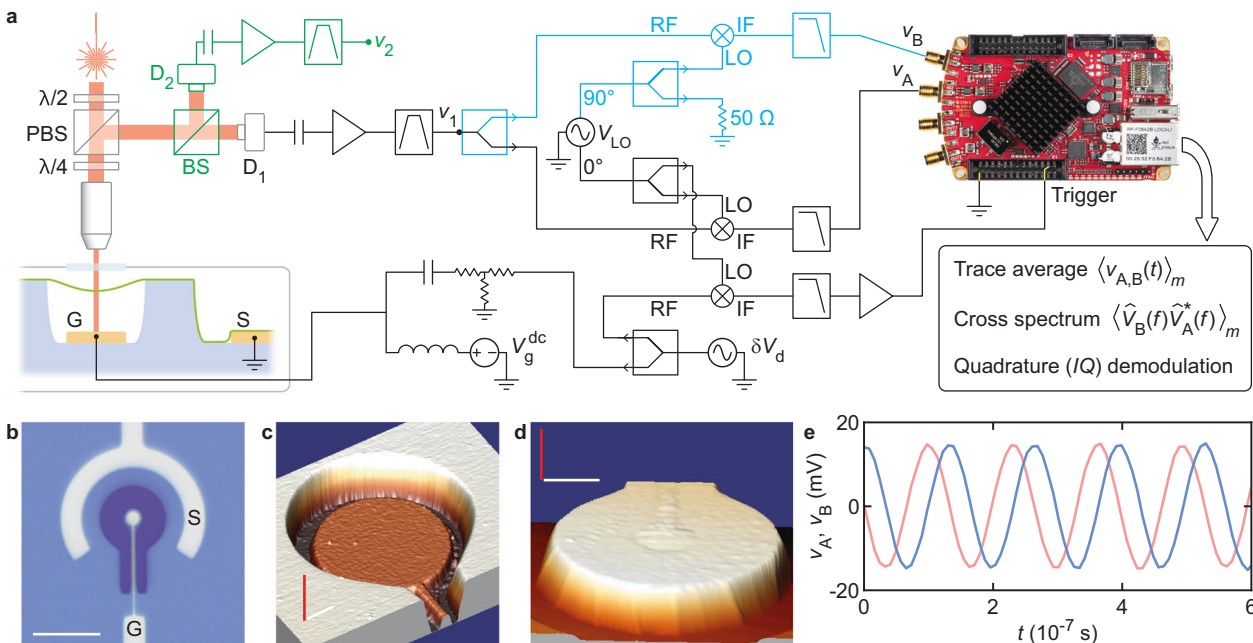

**Fig. 1 | Data acquisition system, measurement setup and resonator. a** The resonator is in vacuum and at room temperature. G, S: gate, source electrodes. $D_1$, $D_2$: avalanche photodetectors. PBS: polarizing beam splitter. BS: 50/50 beam splitter. $\lambda/2$, $\lambda/4$: half-wave, quarter-wave plates. Vibrations are driven by $\delta V_d$. The output signal of $D_1$ is down-converted using a mixer, a local oscillator $V_{LO}$, and a low-pass filter. The down-converted signal is sampled at the receiver (circuit branch in black). Its in-phase $I$ and quadrature $Q$ components can be extracted in software. Alternatively, $I$ and $Q$ can be measured using analog heterodyning by splitting the signal at $v_1$ (circuit branch in blue). Cross-spectra of incoherent signals driven by noise are measured using the branch in black together with the branch in green. **b** Optical image of the substrate hosting the resonator. The dark blue structure is a raised mesa. Scale bar: 15 μm. **c** Atomic force microscope (AFM) image of the cavity in the mesa and of the gate. Scale bars: 600 nm (white, in-plane), 320 nm (red, out-of-plane — height of the mesa). **d** AFM image of FLG covering another mesa in the same way as a tablecloth. Scale bars: 3 μm (white, in-plane), 330 nm (red, out-of-plane —height of the mesa). **e** Voltages $v_A$ and $v_B$ are related to $I$ (red) and $Q$ (blue) down-converted to 7.6 MHz (drive power is − 20 dBm at the gate).

to-digital converters (ADCs) matched to 50 Ohms and with a low phase noise clock providing a sampling rate $f_s = 122.88$ MHz. The sampling rate is less than twice the resonant frequency of the vibrational mode under study, $f_m > 70$ MHz, so the measured signal associated with vibrations cannot be oversampled in the first Nyquist zone and has to be down-converted. This is done with a double-balanced diode ring mixer and a local oscillator at frequency $f_d + f_{trigger}$, where $f_d$ is the frequency of the force that drives vibrations and $f_{trigger} \simeq 1$–$10$ MHz is the frequency of the down-converted signal (for a list of notations, see Supplementary Note 1). The lower frequency part of the response of the mixer at $f_{trigger}$ is measured by the receiver. The driving source, the local oscillator and the source used to trigger the acquisition at $f_{trigger}$ all share the same clock (they share the same frequency reference). The results shown below are obtained in the case where the receiver's ADCs are driven by a separate clock. In Methods, 'Synchronizing the clocks of instruments', we present spectra of vibrational amplitude and vibrational phase measured in configurations where the receiver and the other instruments share the same clock. At room temperature, where the linewidths of vibrational spectra are large, possible offsets between clocks do not adversely affect measurements. At cryogenic temperature, however, where vibrational spectra are narrow, clock synchronization is essential.

We briefly describe our resonator and our vibration detection setup below. FLG is composed of 3 to 4 graphene layers (Supplementary Note 2). The resonator is shaped as a drum and is suspended over a local gate electrode, Fig. 1b–d (Methods, 'Fabrication'). It is kept in vacuum and is measured at room temperature. Its vibrations are driven electrostatically (Methods, 'Actuation of vibrations') and are detected optically (Methods, 'Optical detection of vibrations'). The resonator is placed in an optical standing wave formed between the gate and the surface of a quarter-wave plate facing the resonator (Supplementary Note 3). The optical power incident on the resonator is $\simeq 3\ \mu$W, two to three orders of magnitude smaller than powers used in some other experiments with 2-D resonators[32]. Vibrations modulate the amount of optical energy that the resonator absorbs, resulting in a modulation of the optical intensity measured by a photodetector in the returning light path. The output voltage of the photodetector, $V_{pd}(t)$, oscillates as a function of time at $f_d$ and its amplitude is proportional to the amplitude of vibrations $\delta z_m$. The down-converted copy of $V_{pd}(t)$ is processed in software: it can be averaged in the time domain, its spectrum can be computed, its in-phase and quadrature components can be extracted (Fig. 1e), and information encoded in it as a baseband signal can be retrieved. We demonstrate all these functions below.

### Three estimators for the coherently driven response

The simplest way to characterize the driven response of the resonator is to measure the power of $\delta z_m$ as a function of $f_d$. To this end, we measure the voltage $v_A(t)$ across the input impedance of one input port of the receiver (Fig. 1a). Circuit branches in green and in blue in Fig. 1a are disconnected. $v_A$ is digitized as $v_A(n/f_s)$, $1 \le n \le N$, with $N = 2^{14}$ the number of samples in the digitized trace. The Fourier transform of $v_A$ is computed as $\hat{V}_A(k) = \sum_{n=1}^{N} v_A(n) \exp[-j2\pi(k-1)(n-1)/N]$, where $1 \le k \le N$ and $kf_s/N$ is a Fourier frequency of the receiver. We tune $f_{trigger}$ so it coincides with such a Fourier frequency, $f_{trigger} = kf_s/N$. Using this approach, the computed power spectrum of a single tone $v_A$ as a function of Fourier frequency is a sharp peak at $f_{trigger}$. If $f_{trigger}$ sits between two Fourier frequencies, the power spectrum exhibits spectral leakage but the area under the spectrum remains unaffected. We choose $k = 134$, so the signal at any drive frequency $f_d$ at the output of the photodetector is shifted down to $f_{trigger} \simeq 1$ MHz at the input of the receiver, and we monitor both the peak value of the power spectrum and its area.

Our acquisition system enables separating coherent and incoherent signals in a simple way. After filtering, $v_A$ oscillates coherently at $f_{trigger}$ with an amplitude $\propto \delta z_m$; superimposed are incoherent voltage fluctuations that mostly originate from fluctuations at the output of the photodetector and from voltage noise within the amplifier. The coherent and incoherent parts can be separated thanks to the phase coherence of the acquisition: using the

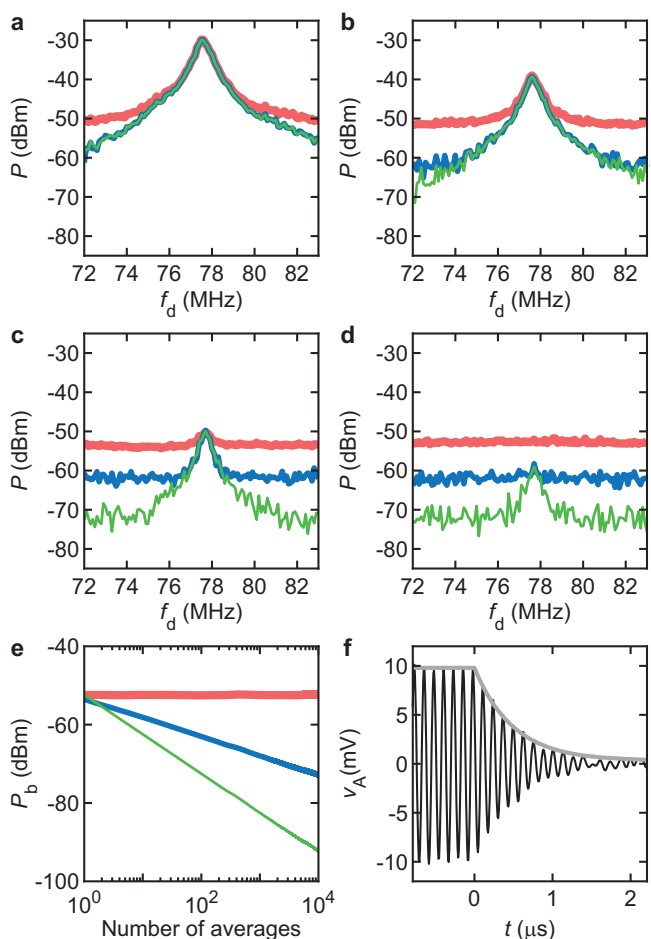

**Fig. 2 | Three power estimators for the driven response. a-d** Average of $m = 100$ realizations of power of voltage $v_A$, $\langle P_{AA} \rangle_m$ (thick red traces); magnitude of the average of $m$ realizations of cross-power of $v_A$, $|\langle P_{AA'} \rangle_m|$ (blue traces); power of the average of $m$ realizations of $v_A$, $\bar{P}_{AA}$ (thin green traces). Drive powers at the gate are $P_d = -40$ dBm (**a**), $-50$ dBm (**b**), $-60$ dBm (**c**), $-70$ dBm (**d**). The drive frequency is $f_d$. **e** Power of noise background $P_b$ away from resonance as a function of number of averages $m$ for $\langle P_{AA} \rangle_m$ (thick red trace), $|\langle P_{AA'} \rangle_m|$ (blue trace), and $\bar{P}_{AA}$ (thin green trace). **f** Ringdown of $v_A$ at $P_d = -40$ dBm measured at $f_{trigger} \simeq 8.2$ MHz as a function of time $t$ (black trace). The envelope in gray is a fit to an exponential decay in a linear damping model. $V_g^{dc} = -15$ V in all panels.

coherent drive signal to trigger the acquisition, the coherent parts in the voltage waveforms $v_A$ received sequentially are all in phase with each other while the incoherent parts can be averaged away. This simple (and usual) process is illustrated in Fig. 2a–d using three estimators for the power of $v_A$ as a function of $f_d$. Here, 4 different drive powers $P_d$ are used (from a to d: $P_d = -40$ dBm, $-50$ dBm, $-60$ dBm, $-70$ dBm). Red traces show the averaged power $\langle P_{AA} \rangle_m = \langle 2|\hat{V}_A(k = 134)|^2/N^2 \rangle_m$, where $\langle \cdot \rangle_m$ denotes an average over $m = 100$ realizations. Blue traces show the averaged cross-power $|\langle P_{AA'} \rangle_m| = |\langle 2\hat{V}_A(k = 134)\hat{V}_{A'}^*(k = 134)/N^2 \rangle_m|$, where $\hat{V}_{A'}$ is the transform of a separate realization of $v_A$. While cross-power measurements usually require a two-channel instrument to acquire two signals in parallel, here we compute the cross-power of pairs of $v_A$ acquired sequentially, given that the photodetector noise and the amplifier noise are stationary and ergodic processes that are statistically independent of each other (we also assume that thermomechanical noise is weak compared to other sources of noise). Green traces show the power of the average of $m$ realizations of $v_A$, that is, the power computed after averaging away some of the incoherent part. It reads $\bar{P}_{AA} = 2|\hat{W}_m(k = 134)|^2/N^2$, with $\hat{W}_m$ the transform of $\langle v_A \rangle_m$. All power estimators are single sided. The code to compute the power estimators is shown in Supplementary Note 4.

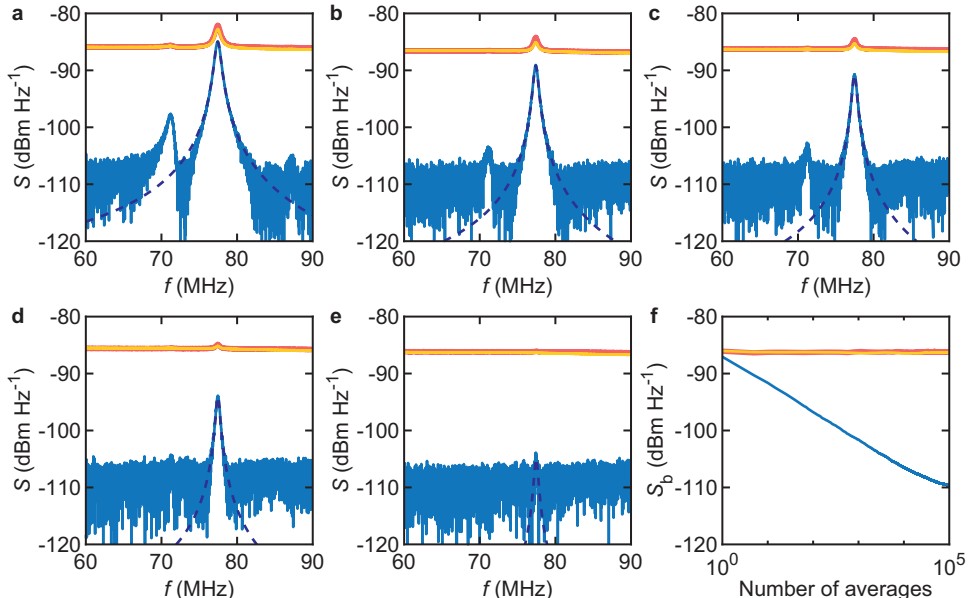

**Fig. 3 | Power spectra of displacement fluctuations induced by a force noise.**
**a–e** Averaged power spectral densities $\langle S_{v_1 v_1} \rangle_m$ (red) and $\langle S_{v_2 v_2} \rangle_m$ (orange) and averaged cross-power spectral density $|\langle S_{v_2 v_1} \rangle_m|$ (blue) as a function of Fourier frequency for various voltage noise intensities $S_{uu}$, with $m = 10^5$. From (**a–e**): $S_{uu} \simeq 5.9 \times 10^{-12}$ $V_{rms}^2$ $Hz^{-1}$, $T_{eff} \simeq 1.2 \times 10^6$ K (**a**); $S_{uu} \simeq 2.9 \times 10^{-12}$ $V_{rms}^2$ $Hz^{-1}$, $T_{eff} \simeq 7.6 \times 10^5$ K (**b**); $S_{uu} \simeq 1.4 \times 10^{-12}$ $V_{rms}^2$ $Hz^{-1}$, $T_{eff} \simeq 4.1 \times 10^5$ K (**c**); $S_{uu} \simeq 5.2 \times 10^{-13}$ $V_{rms}^2$ $Hz^{-1}$,

$T_{eff} \simeq 1.7 \times 10^5$ K (**d**); $S_{uu} \simeq 6 \times 10^{-14}$ $V_{rms}^2$ $Hz^{-1}$, $T_{eff} \simeq 2 \times 10^4$ K (**e**). $V_g^{dc} = -15$ V. Dashed traces are fits to Lorentzian lineshapes. **f** Off-resonance background $S_b$ as a function of number of averages $m$ for $\langle S_{v_1 v_1} \rangle_m$ (red), $\langle S_{v_2 v_2} \rangle_m$ (orange) and $|\langle S_{v_2 v_1} \rangle_m|$ (blue).

The freedom of data processing offered by our SDR approach allows us to identify the best estimator for the driven response. At large drive, all power estimators as a function of $f_d$ exhibit a similar Lorentzian resonance (Fig. 2a). As $P_d$ decreases, the response estimated from $\langle P_{AA} \rangle_m$, which is the default mode in Fast Fourier Transform analyzers, quickly flattens out. What is left is a noise background whose power is $2\mathbb{V}\{\delta u\}/N$, where $\mathbb{V}\{\delta u\}$ is the variance of voltage fluctuations from the photodetector and the amplifier. Better estimators are based on the averaged cross-power and on the power of coherently averaged signals. Unlike the former estimator, the power of the background $P_b$ of the latter two estimators decreases as the number of averages $m$ increases. As shown in Fig. 2e, multiplying the number of averages by a factor of 10 results in a decrease of $P_b$ by 5 dB for $|\langle P_{AA'} \rangle_m|$ and by 10 dB for $\bar{P}_{AA}$[33,34]. Weak $P_b$ and correspondingly high signal-to-noise ratio can be obtained if large $m$ can be afforded. This entails that the whole measurement setup must not drift in the course of long averaging processes.

A good estimator for the driven response is important to measure the quality factor $Q_m$ of the vibrational mode. The latter can be estimated from the response in frequency, $Q_m = f_m/\Delta f$, where $\Delta f$ is the full width at half maximum of the power response. However, nondissipative spectral broadening due to resonant frequency fluctuations[35–44] may result in an underestimate of $Q_m$. Ringdown experiments, where the drive is suddenly switched off and vibrations are left to freely decay, offer an unambiguous estimation of $Q_m$ as they solely measure energy[45–49]. By integrating a radio frequency switch in our drive circuit (Methods, 'Detailed measurement circuits'), we performed such ringdown experiments. As shown in Fig. 2f, the amplitude of vibrations down-converted to 8.2 MHz and averaged $2 \times 10^4$ times decays as $\sim \exp[-t/(2\tau)]$ with $\tau = Q_m/(2\pi f_m)$. Using $f_m = 77.6$ MHz we find $Q_m \simeq 120$, which coincides with the estimate based on the resonance linewidth (Fig. 2a). This result agrees with earlier ringdown measurements in $MoS_2$ resonators, which showed that at room temperature the linewidth of the resonance is mostly accounted for by energy dissipation, with no visible contribution from frequency fluctuations[46]. We also verified that the averaging process does not affect our estimation of $\tau$. Indeed, phase and frequency noise in the signal would lower the amplitude of the averaged

driven signal but would not modify the averaged exponential decay. Further, we measured the time jitter between subsequent trigger events and found it to be at most 10 ns; our numerical calculations show that this time jitter does not affect the averaged exponential decay since $\tau \simeq 250$ ns.

## Power spectra of displacement fluctuations induced by a force noise

Our acquisition system enables measuring the spectral response of the resonator to a force noise[50–52]. Here, vibrations are incoherent, so the one-channel acquisition trick used with coherently driven vibrations will not work and a two-channel acquisition is needed. Such is the purpose of the circuit branch in green in Fig. 1a, which involves a second photodetector and a second amplifier whose characteristics are as close as possible to those of the detector and of the amplifier in the branch in black (the branch in blue is still disconnected). To minimize the phase offset between the two parallel measurements, the path from BS to $v_1$ and the path from BS to $v_2$ in Fig. 1a have the same length. Finally, the circuit to the right of node $v_1$ in Fig. 1a is disconnected and is replaced by a digital oscilloscope with a sampling rate set to 5 GHz. In doing so, we avoid the problem of duplicating the circuit dedicated to the down-conversion to $f_{trigger}$, which may introduce phase and amplitude imbalances. Vibrations are driven incoherently by an electrostatic force noise $S_{FF} = 4S_{uu}(C_g' V_g^{dc})^2$ created by applying a calibrated voltage noise $4S_{uu}$ to the gate. $S_{uu}$ is first measured across the 50 Ohm input impedance of a spectrum analyzer and the factor of 4 accounts for full reflection at the gate. $C_g' \simeq 8 \times 10^{-10}$ F $m^{-1}$, the gradient of gate capacitance in the flexural direction, is estimated from the geometry of the device and using COMSOL. $V_g^{dc}$ is the dc voltage between the gate and the resonator. From the equipartition theorem, $S_{FF}$ can be represented by an effective modal temperature $T_{eff} = S_{FF}/(8\pi k_B m_{eff} f_m/Q_m)$ indicated in the captions to Fig. 3. The voltage measured at node $v_1$ is $v_1(t) = c(t) + a(t)$, where $c(t)$ represents voltage fluctuations transduced from displacement fluctuations and $a(t)$ represents voltage fluctuations from the photodetector and the amplifier. Similarly, $v_2(t) = c(t) + b(t)$, where $b(t)$ represents nonmechanical voltage fluctuations from the other branch. We compute the averaged single-sided power spectral density of voltage fluctuations for each node

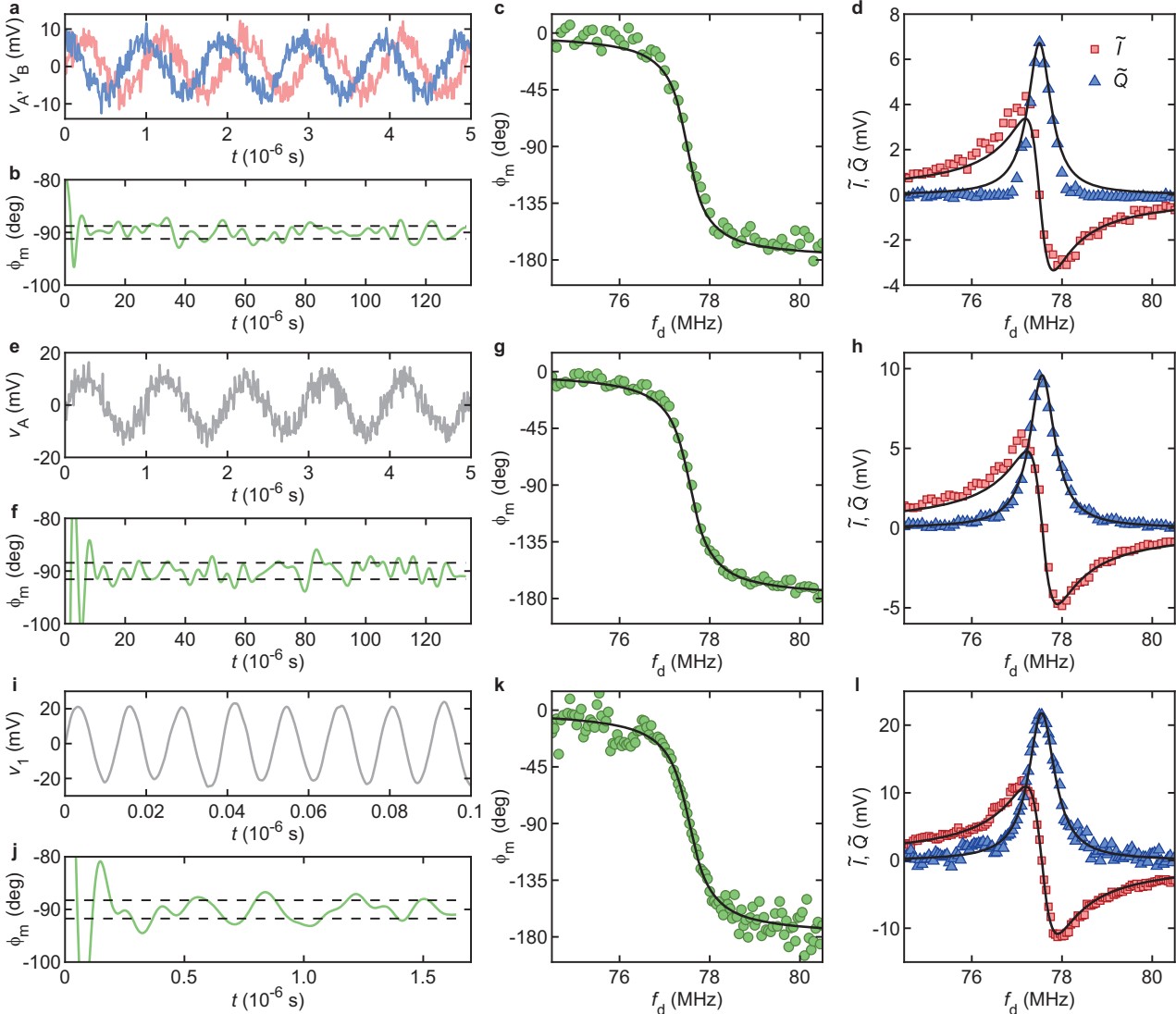

**Fig. 4 | Measuring the in-phase $\tilde{I}$ and quadrature $\tilde{Q}$ components of down-converted vibrations as a function of drive frequency $f_d$. a–d** Approach (i) based on analog demodulation with SDRlab using two mixers and two local oscillators in quadrature. **e–h** Approach (ii) based on digital demodulation with SDRlab using one mixer and one local oscillator and $IQ$ demodulation in software. **i–l** Approach (iii)

based on digital demodulation with an oscilloscope and $IQ$ demodulation in software. $P_d = -40$ dBm, $V_g^{dc} = -15$ V. Black traces are fits to a harmonic oscillator response. Dashed lines in (**b**, **f**, **j**) indicate the standard deviation of vibrational phase $\phi_m$ at resonance. $v_A$ and $v_B$ in (**a**) and (**e**) and $v_1$ in (**i**) refer to voltages at the nodes shown in Fig. 1a.

(Supplementary Note 4). At node $v_1$, it reads $\langle S_{v_1 v_1} \rangle_m = 2[\langle |\hat{C}|^2 \rangle_m + \langle |\hat{A}|^2 \rangle_m + \langle \hat{C}\hat{A}^* \rangle_m + \langle \hat{A}\hat{C}^* \rangle_m]/(Nf_s) \rightarrow \langle S_{cc} \rangle_m + \langle S_{aa} \rangle_m$. We also compute the averaged single-sided cross-power spectral density. Its magnitude reads $|\langle S_{v_2 v_1} \rangle_m| = 2|\langle |\hat{C}|^2 \rangle_m + \langle \hat{C}\hat{B}^* \rangle_m + \langle \hat{A}\hat{C}^* \rangle_m + \langle \hat{A}\hat{B}^* \rangle_m|/(Nf_s) = S_{cc} + O(1/\sqrt{m})$. Figures 3a–e display $\langle S_{v_1 v_1} \rangle_m$ (red), $\langle S_{v_2 v_2} \rangle_m$ (orange), and $|\langle S_{v_2 v_1} \rangle_m|$ (blue) as a function of Fourier frequency using $m = 10^5$. $T_{eff}$ decreases from a to e. Once again, $\langle S_{v_1 v_1} \rangle_m$ and $\langle S_{v_2 v_2} \rangle_m$ quickly flatten out to $2\mathbb{V}\{a\}/f_s$ and $2\mathbb{V}\{b\}/f_s$, respectively. In stark contrast, the effect of averaging photodetector noise and amplifier noise in $|\langle S_{v_2 v_1} \rangle_m|$ (Fig. 3f) means that the mechanical response can be resolved in the cross-power spectrum down to the lowest force noise (simulations of the Brownian motion of the resonator are consistent with our measurements, see Supplementary Note 5). This allows us to extract the full width at half maximum $\Delta f$ from a fit of $|\langle S_{v_2 v_1} \rangle_m|$ to a Lorentzian lineshape, which reveals that $\Delta f$ increases with $S_{uu}$. Because the resonant frequency does not noticeably shift as $S_{uu}$ increases, this lineshape broadening may not be caused by frequency fluctuations nonlinearly transduced by displacement fluctuations[50,53] (unless the resonator is in a zero-dispersion regime[51]). Instead, we find that $\Delta f(S_{uu})$ is

well accounted for by a nonlinear damping model[54,55] (Supplementary Note 5).

## In-phase and quadrature components of vibrations

Our system can emulate a vector network analyzer, allowing us to measure the in-phase $I$ and quadrature $Q$ components of vibrations. Such vector measurements can be done with a directional coupler to separate transmitted and reflected powers. Instead, here we adopt a heterodyne $I$ and $Q$ demodulation technique. We compare three different approaches: (i) analog demodulation with SDRlab using two mixers and two local oscillators in quadrature; (ii) digital demodulation with SDRlab using one mixer and one local oscillator for down-conversion and $IQ$ demodulation in software; (iii) digital demodulation with an oscilloscope without down-conversion and with $IQ$ demodulation in software. We find that (ii) yields $I$ and $Q$ spectra that most resemble the frequency response of a harmonic oscillator.

Below, we briefly describe our simple measurement technique used in approach (i), which is depicted in Fig. 1a (black and blue circuit branches). Approaches (ii) and (iii) are similar and simpler. We assume a

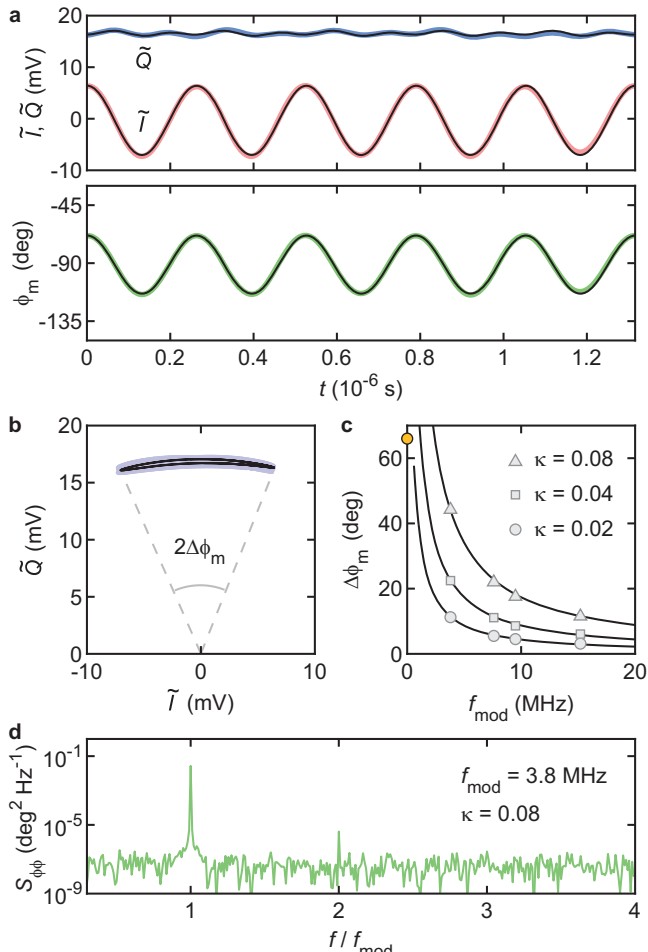

**Fig. 5 | Phase modulation induced by strain modulation. a** In-phase and quadrature components $\tilde{I}$ and $\tilde{Q}$ of the down-converted photodetector signal (top panel) and demodulated vibrational phase $\phi_m$ with respect to drive (bottom panel) as a function of time. Modulation strength and modulation frequency are $\kappa = 0.04$ and $f_{mod} = 3.8$ MHz, respectively. **b** Peak amplitude of phase modulation $\Delta\phi_m$ represented in $(\tilde{I}, \tilde{Q})$ space. **c** $\Delta\phi_m$ as a function of $f_{mod}$ for various values of $\kappa$. The orange dot shows $\Delta\phi_m$ for $\kappa = 0.02$ and $f_{mod} \simeq 10^3$ Hz. Black traces in (**a**–**c**) are calculations based on a linear equation of motion including a modulated spring constant. **d** Power spectral density of $\phi_m$ for $\kappa = 0.08$ and $f_{mod} = 3.8$ MHz. $P_d = -40$ dBm and $V_g^{dc} = -14.7$ V in all panels.

photodetector output voltage of the form $V_{pd}(t) = V_{pd}\cos(2\pi f_d t + \phi_m) = I\cos(2\pi f_d t) + Q\sin(2\pi f_d t)$, with $I = V_{pd}\cos\phi_m$ and $Q = -V_{pd}\sin\phi_m$. The signal is split at node $v_1$ and down-converted to $v_A$ and $v_B$. The latter read

$$v_A = \frac{\alpha}{2\sqrt{2}}\left[I\cos(2\pi f_{trigger}t + \phi') - Q\sin(2\pi f_{trigger}t + \phi')\right] \quad (1)$$

$$v_B = \frac{\alpha}{2\sqrt{2}}\left[I\sin(2\pi f_{trigger}t + \phi') + Q\cos(2\pi f_{trigger}t + \phi')\right], \quad (2)$$

with $\alpha$ an attenuation factor (primarily accounting for mixer insertion loss) and $\phi'$ a phase offset. Note that $v_A^2 + v_B^2 = (I^2 + Q^2)\alpha^2/8$, which means that the measurement of the vibrational energy does not depend on the demodulation procedure (within an attenuation factor) if $\phi'$ is the same for both $v_A$ and $v_B$ and if the two local oscillators are at the same frequency. $v_A$ and $v_B$ are digitized, and 500 time traces are acquired and averaged (Fig. 4a). Averaged $v_A$ and $v_B$ are down-converted to dc in software by multiplying

each of them by $\cos(2\pi\tilde{f}_{trigger}t + \phi'')$ and low-pass filtering the products. $\tilde{f}_{trigger}$ and $\phi''$ are both adjusted in software. $\tilde{f}_{trigger}$ is close to $f_{trigger}$, with a small adjustable offset that comes from the fact that SDRlab has its own frequency reference. The results of the down-conversion of $v_A$ and $v_B$ to dc are labeled as $v_a$ and $v_b$, respectively. They read

$$v_a = \frac{\alpha}{4\sqrt{2}}\left[I\cos(\phi' - \phi'') - Q\sin(\phi' - \phi'')\right] \quad (3)$$

$$v_b = \frac{\alpha}{4\sqrt{2}}\left[I\sin(\phi' - \phi'') + Q\cos(\phi' - \phi'')\right]. \quad (4)$$

$\phi''$ is adjusted in such a way that, on resonance ($f_d \simeq f_m$) where $I = 0$, $v_a \simeq 0$ and $\arctan(v_b/v_a) = \phi_m \simeq -\pi/2$ (Fig. 4b). Away from resonance, $f_d \neq f_m$, we find that $\arctan(v_b/v_a)$ changes with $f_d$ linearly owing to propagation delay through the cables. We subtract this linear dependence from $\arctan(v_b/v_a)$ (Methods, 'In-phase and quadrature components of vibrations') and obtain $\tilde{\phi}_m(f_d)$ (Fig. 4c). We obtain the down-converted in-phase $\tilde{I}$ and quadrature $\tilde{Q}$ components of the photodetector output signal as $\tilde{I} = 2(v_a^2 + v_b^2)^{1/2}\cos(\phi_m)$ and $\tilde{Q} = -2(v_a^2 + v_b^2)^{1/2}\sin(\phi_m)$. Results shown in Fig. 4d resemble the response of a harmonic oscillator, $\tilde{I} \propto (f_m^2 - f_d^2)/D$, $\tilde{Q} \propto (f_m f_d/Q_m)/D$, $D = (f_m^2 - f_d^2)^2 + (f_m f_d/Q_m)^2$. We suspect that deviations from the model are caused by slight phase and amplitude imbalances between the two local oscillators, which result in $v_A^2 + v_B^2 \neq (I^2 + Q^2)\alpha^2/8$ and in distorted $\tilde{I}(f_d)$, $\tilde{Q}(f_d)$.

To circumvent this problem, we use a single mixer for down-conversion to $v_A$ and demodulate $v_A$ in software (approach (ii)). The circuit in blue in Fig. 1a is disconnected; because no splitter is involved, the amplitude of $v_A$ is enlarged by $\sqrt{2}$ (Fig. 4e). $v_A$ is then multiplied by $\cos(2\pi\tilde{f}_{trigger}t + \phi'')$ and by $\sin(2\pi\tilde{f}_{trigger}t + \phi'')$ in software. The products are low-pass filtered, yielding $v_a$ and $v_b$. The resulting $\phi_m$ is shown in Fig. 4f, g. $\tilde{I}(f_d)$ and $\tilde{Q}(f_d)$, shown in Fig. 4h, are close to the response of a harmonic oscillator. We note that our technique of demodulating $I$ and $Q$ in software and the previous technique based on two mixers have a similar measurement bandwidth. This is because the software technique still requires a mixer for down-conversion, and because in both cases most of the measurement time is spent downloading data to the host computer and averaging them. Overall, measuring averaged $I$ and $Q$ spectra with either technique requires the same amount of time.

Finally, we do away with mixer insertion loss (approach (iii)). The circuit to the right of node $v_1$ is disconnected and is replaced with an oscilloscope. Compared to single channel $v_A$, $v_1$ is enlarged by $\simeq 1/\alpha$ (Fig. 4i). In addition, 4096 time traces are acquired and averaged by the oscilloscope, compared with 500 averages with SDRlab (which may explain why $v_1(t)$ in Fig. 4i displays a lesser amount of noise than $v_A(t)$ in Fig. 4e). $v_1$ is then multiplied by $\cos(2\pi f_d t + \phi'')$ and by $\sin(2\pi f_d t + \phi'')$ in software. After low-pass filtering of the products, $\phi_m$ (Fig. 4j, k) and $\tilde{I}(f_d)$ and $\tilde{Q}(f_d)$ (Fig. 4l) are obtained. Approaches (ii) and (iii) yield similar results, with (ii) yielding somewhat cleaner data.

## Phase modulation induced by strain modulation

Not only can we measure the phase of vibrations $\phi_m$ with respect to the drive, we can also control it through strain modulation. 2-D resonators are well suited to this application because their resonant frequencies are highly tunable[5]. Modulating $\phi_m$ by modulating strain within the resonator is straightforward. The process goes as follows. Near resonance, $\phi_m$ changes linearly with drive frequency $f_d$, unlike the amplitude of vibrations that changes only weakly (see, e.g., Fig. 4g). Weakly modulating strain induces a weak modulation of the resonant frequency $f_m$[27]. The effect on $\phi_m$ resembles the effect of modulating $f_d$ with $f_m$ fixed: $\phi_m$ gets modulated at the frequency of strain modulation. Strain modulation is conveniently used to parametrically amplify the amplitude of vibrations[56,57]. It is also used to resonantly couple vibrational modes that are distant in frequency[58], resulting in an energy exchange that may cool down or heat up vibrations[59,60].

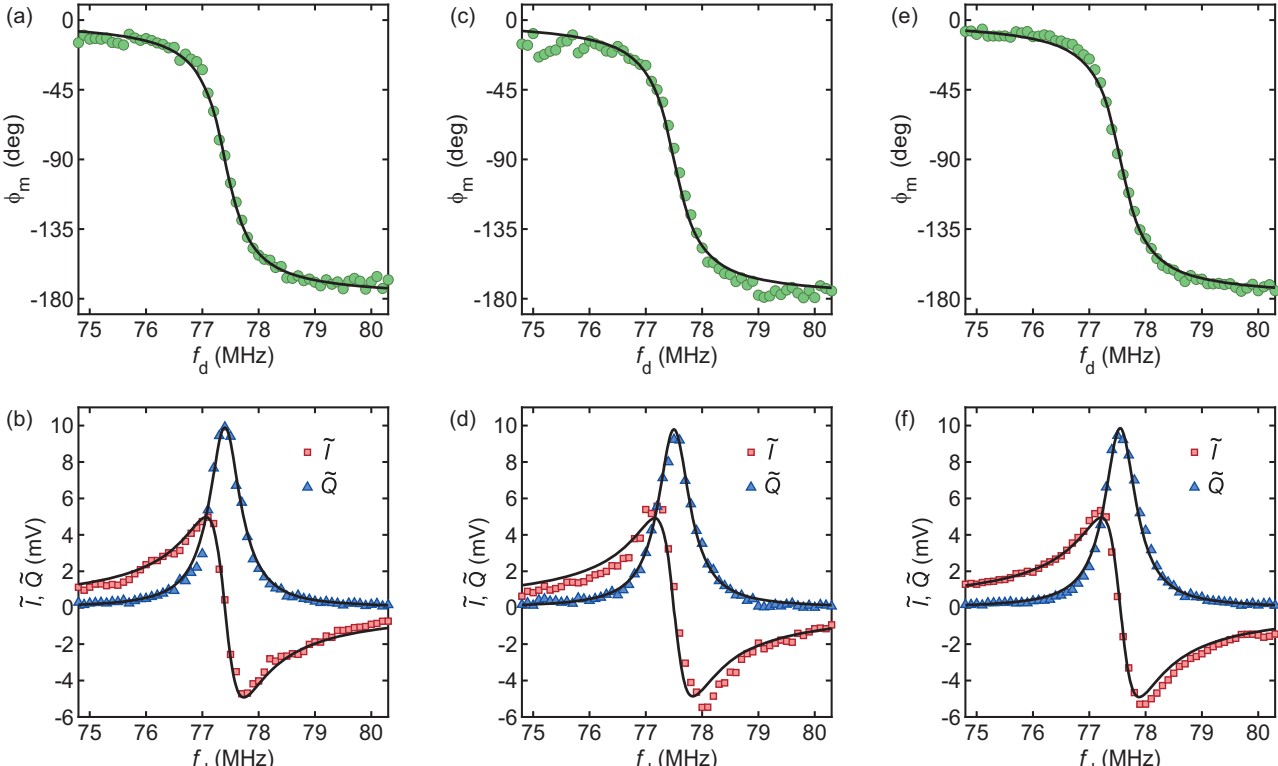

**Fig. 6 | Synchronizing the clocks of instruments. a**, **c**, **e** Demodulated vibrational phase $\phi_m$ with respect to drive as a function of drive frequency $f_d$. **b**, **d**, **f** In-phase and quadrature components $\tilde{I}$ and $\tilde{Q}$ of the down-converted photodetector signal as a function of $f_d$. In (**a**, **b**), a 0 dBm, 10 MHz signal at the output of SDRLab is used as a clock for all instruments. In (**c**, **d**), SDRlab 122-16 External Clock Standard Kit is used with the clock signal supplied by a signal generator from Rohde & Schwarz. In (**e**, **f**), the standard SDRlab receiver running on its own clock is used.

Here, we measure $\phi_m$ in the time domain in response to a modulated strain with various modulation frequencies and modulation strengths. To our knowledge, such a study has not previously been carried out. In addition to a drive near resonance, we apply a second drive at a lower frequency $f_{mod} < f_m$. This drive modulates the spring constant as $k_m \rightarrow k_m[1 - \kappa\cos(2\pi f_{mod}t)]$ with $\kappa \ll 1$, resulting in a frequency modulation $f_m \rightarrow f_m[1 - (\kappa/2)\cos(2\pi f_{mod}t)]$. The modulation strength reads $\kappa = (2/f_m)(\partial f_m/\partial V_g^{dc})\delta V_{mod}$, where $\delta V_{mod}$ is the amplitude of the modulation voltage. We demodulate $\phi_m$ both with SDRlab (approach (i) from the previous section) and with the oscilloscope (approach (iii)) and find similar results. Figure 5a shows $\tilde{I}$, $\tilde{Q}$ and $\phi_m$ as a function of time for $\kappa = 0.04$, $f_{mod} = 3.8$ MHz, and $f_m \simeq 76$ MHz. $\tilde{I}$ and $\phi_m$ oscillate at $f_{mod}$, while $\tilde{Q}$ is only weakly modulated. Figure 5b shows the peak amplitude of phase modulation $\Delta\phi_m$ in $(\tilde{I}, \tilde{Q})$ space. Interestingly, we find that $\Delta\phi_m$ decreases with increasing $f_{mod}$ (Fig. 5c). We numerically solve a linear equation of motion for our resonator that includes a modulated spring constant of the above form and succeed in reproducing $\tilde{I}(t)$, $\tilde{Q}(t)$, $\phi_m(t)$ (black traces in Fig. 5a, b) and $\Delta\phi_m(f_{mod}, \kappa)$ (black traces in Fig. 5c) without fit parameters. Figure 5d shows the power spectral density of $\phi_m$, $S_{\phi\phi}$, for $\kappa = 0.08$ and $f_{mod} = 3.8$ MHz. Even at this large modulation strength and this low modulation frequency, $\phi_m$ still oscillates harmonically and mostly as a single tone. Details of the dynamics of $\tilde{I}$ and $\tilde{Q}$ for various values of $\kappa$ and $f_{mod}$ are shown in Supplementary Note 6.

## Limitations of the study and conclusion

Our data acquisition system, based on a SDR architecture, allows us to perform a variety of nanomechanical measurements that would normally require the use of a range of different instruments. We measure the spectrum and the ringdown of coherently driven vibrations. We measure the cross-power spectrum of coherently and incoherently driven vibrations. We perform vector measurements of the in-phase and quadrature components of vibrations. We control the vibrational phase and modulate it at high frequency. Our approach allows us to tailor our experiments at will and gives us control over every stage of data processing. We find that it facilitates the teaching of radio wave engineering and physics, and is a convenient alternative when more advanced radio frequency equipment is not available. A limitation of our system is the low frequency resolution of the measured spectra, $f_s/(2N) = 3750$ Hz with $N = 2^{14}$. It can be easily improved by downsampling the down-converted signal. For example, using a lower sampling rate $f_s/8 = 15.36$ MHz, a signal down-converted to ~1 MHz is still oversampled in the first Nyquist zone while the frequency resolution is 470 Hz. Furthermore, by properly band-pass filtering the signal around $f_m$ and sampling the band-limited signal in a higher order Nyquist zone (undersampling), it may be possible to both improve the frequency resolution and to use the ADC of the receiver as a down-converter. Another limitation is the rather low throughput of our system. Much time is spent downloading data from the radio receiver and processing them with the host computer. This requires that the non-SDR part of the experimental setup, including laser, vacuum chamber, and cables, be as stable as possible over the course of several hours. Long averaging processes are especially challenging because they may be spoiled by laser power drift and temperature drift in the laboratory. Whenever such drifts were observed, measurement data were systematically discarded. To solve this issue, we are currently investigating the use of on-board data processing in FPGA. While this task may be best suited to systems such as PCI extensions for instrumentation (PXI), where PCI stands for Peripheral Component Interconnect, their high cost is an issue. Instead, we are investigating the use of more affordable systems, such as FPGA

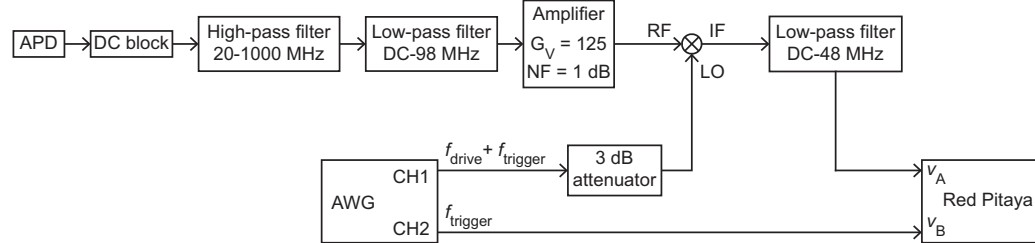

**Fig. 7 | Setup for measuring the cross-power spectrum of driven vibrations.** APD: avalanche photodetector. AWG: arbitrary waveform generator. CH1, CH2: output channels 1 and 2 of AWG. The voltage gain and the noise figure of the amplifier are denoted by $G_V$ and NF, respectively. RF, LO and IF stand for radio frequency input signal, local oscillator and intermediary frequency output signal, respectively. $f_{drive}$ and $f_{trigger}$ denote the frequency of the driving signal and the frequency of the trigger signal, respectively.

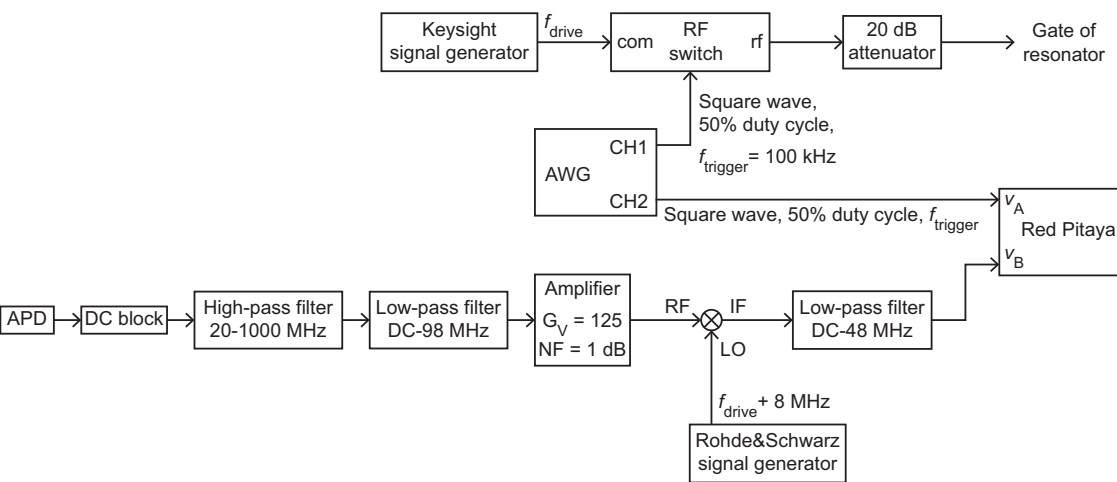

**Fig. 8 | Setup for measuring the vibrational ringdown.** APD: avalanche photodetector. AWG: arbitrary waveform generator. CH1, CH2: output channels 1 and 2 of AWG. The voltage gain and the noise figure of the amplifier are denoted by $G_V$ and NF, respectively. RF, LO and IF stand for radio frequency input signal, local oscillator and intermediary frequency output signal, respectively. $f_{drive}$ and $f_{trigger}$ denote the frequency of the driving signal and the frequency of the trigger signal, respectively. RF switch: 'com' port is the RF common input port of the switch, and 'rf' port is the output port of the switch.

development boards, which may enable our full data acquisition process within an embedded platform[61].

## Methods
### Synchronizing the clocks of instruments
Here we address the topic of clock synchronization among the receiver, the driving source, and the local oscillator used for down-conversion. To illustrate the discussion, we present measurements of the vibrational amplitude and of the vibrational phase as a function of drive frequency $f_d$ carried out with 3 different clock configurations. Data are obtained using the technique of demodulating $I$ and $Q$ in software with SDRlab and may be compared with data in Fig. 4g, h. As in Fig. 4, the same device is measured at room temperature, the drive power is $P_d = -40$ dBm, the gate voltage is $V_g^{dc} = -15$ V, the incident optical power is $P_{inc} \simeq 3$ μW, and measured signals are down-converted to ~ 1 MHz and averaged 500 times. However, data in this section were measured several months after those in Fig. 4, after the chip holding the device had undergone several manipulations. Data in Fig. 6a, b are obtained using a 0 dBm, 10 MHz signal at the output of SDRLab that is employed as a clock for all instruments. Data in Fig. 6c, d are obtained using a different type of SDRlab instrument that has to be used with an external clock at 122.8 MHz (SDRlab 122-16 External Clock Standard Kit). To supply the clock signal, we use a low-noise signal generator from Rohde & Schwarz, and connect its 10 MHz frequency reference output port to the 10 MHz frequency reference input ports of the driving source and of the local oscillator. Data in Fig. 6e, f are obtained

with our standard SDRlab receiver running on its own clock, as in Fig. 4g, h. Data shown in Fig. 6a–f are obtained within the same measurement session. As in Fig. 4, solid traces are fits to the response of a harmonic oscillator. We find that the 3 different clock configurations yield qualitatively similar results. At room temperature, where spectral linewidths are large, possible offsets between clocks do not adversely affect measurements. At cryogenic temperature, however, where vibrational spectra are narrow, clock synchronization is essential and would have to be implemented.

### Fabrication
FLG is transferred onto a raised mesa etched in silicon oxide (dark feature in Fig. 1b) and is contacted by a C-shape source electrode. The gate is patterned at the bottom of a cavity etched in the mesa (Fig. 1c). The thickness of the gate is 90 nm to make it reflective in the visible range. The gap size between the resonator and the top of the gate electrode is nominally 230 nm to optimize the optical responsivity of the resonator at wavelength $\lambda = 633$ nm. The optical responsivity reads $(\partial P_{ref}/\partial z)/P_{in} \simeq 3 \times 10^{-3}$ nm$^{-1}$, where $P_{in}$ is the optical power incident on the resonator and $P_{ref}$ is the optical power reflected by the device formed by the resonator and the gate (Supplementary Note 3). FLG covers the mesa in a way reminiscent of a tablecloth (Fig. 1d). This structure was intended to improve the flatness of the resonator and was inspired by the flatness of marmalade jar sealing films. Interestingly, we later learned about an earlier work in which graphene is transferred onto a recess instead of a mesa, presumably for the same purpose[62].

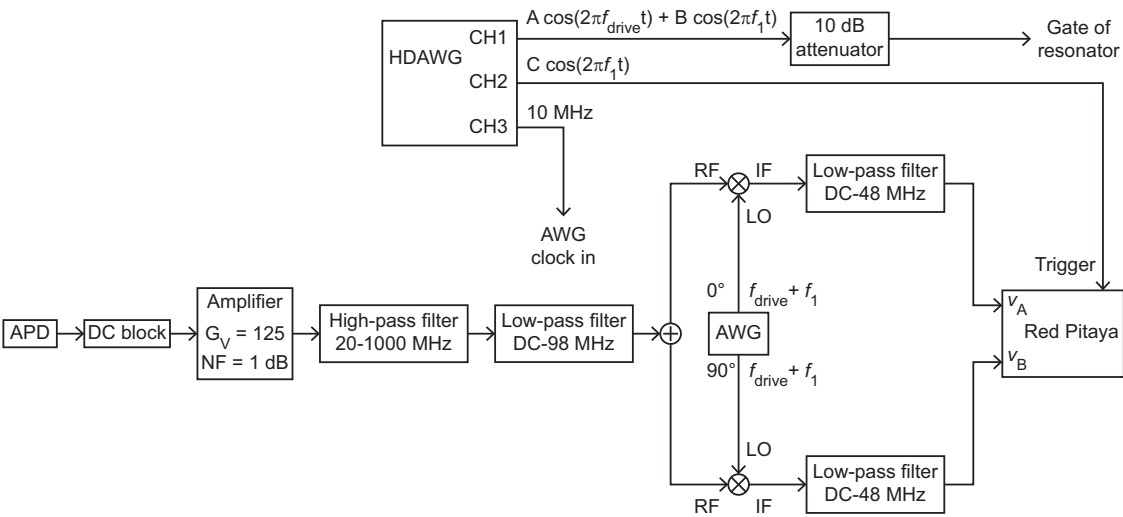

**Fig. 9 | Setup for measuring modulations of the vibrational phase.** APD: avalanche photodetector. AWG: arbitrary waveform generator. The voltage gain and the noise figure of the amplifier are denoted by $G_V$ and NF, respectively. RF, LO and IF stand for radio frequency input signal, local oscillator and intermediary frequency output signal, respectively. HDAWG: high definition arbitrary waveform generator with 750 MHz bandwidth. CH1, CH2, CH3: output channels 1, 2 and 3 of HDAWG. $f_{drive}$ and $f_{trigger}$ denote the frequency of the driving signal and the frequency of the trigger signal, respectively. $f_1$ is the frequency of the output signal at CH2.

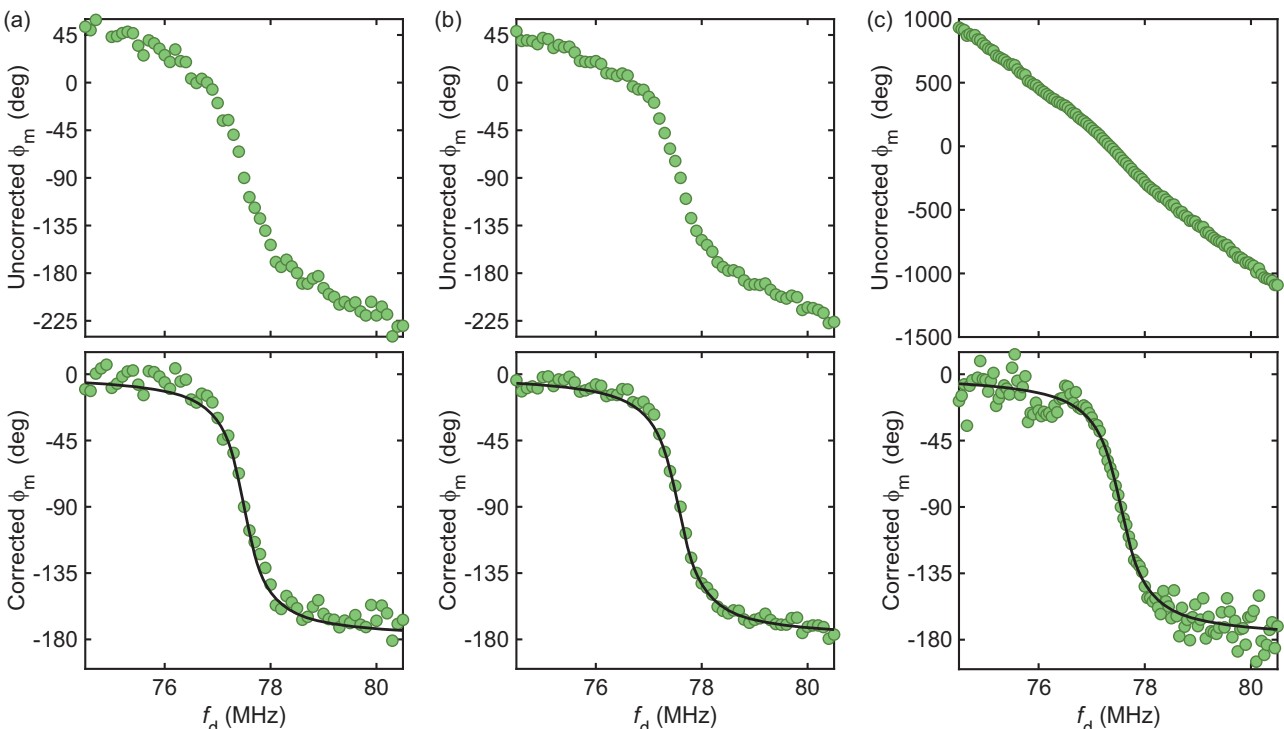

**Fig. 10 | Correcting the vibrational phase $\phi_m$. a** Approach (i). **b** Approach (ii). **c** Approach (iii). Black traces are fits to the response of a harmonic oscillator. $P_d = -40$ dBm, $V_g^{dc} = -15$ V. $f_d$ is the drive frequency.

## Actuation of vibrations

In the simplest case of an unmodulated drive, the driving source outputs a single tone signal at $f_d$ that travels to the gate electrode (the resonator is grounded through its source electrode). This signal drives the resonator with an electrostatic force of rms amplitude $\delta F_d \simeq (1 + \Gamma)C'_g V_g^{dc} \delta V_d$, where $\Gamma \simeq 1$ is the reflection coefficient at the source electrode, $C'_g = dC_g/dz$ is the gradient of gate capacitance in the flexural direction, $V_g^{dc}$ is the dc voltage between the gate and the resonator, and $\delta V_d$ is the rms voltage of the drive signal. In the regime of linear vibrations, the complex amplitude of vibrations at $f_d$ is $\delta z_m = \sqrt{2}\delta F_d \chi / m_{eff}$, with $m_{eff}$ the effective mass of the vibrational mode and $\chi = [4\pi^2(f_m^2 - f_d^2 - jf_m f_d/Q_m)]^{-1}$ the linear response function of the resonator that features the resonant frequency $f_m$ and the quality factor $Q_m$ of the mode.

## Optical detection of vibrations

We use a Helium-Neon laser as an optical source emitting at $\lambda = 633$ nm. We filter the output of the laser with a single mode fiber and control its linear polarization with a half-wave plate. We use a combination of a polarizing

beam splitter and a quarter-wave plate to steer the beam reflected off the device into the photodetector, as follows. Linearly polarized light incident on the surface of the quarter-wave plate facing the polarizing beam splitter (Fig. 1a) becomes circularly polarized upon traversing the quarter-wave plate. It then propagates to the device where it gets reflected by the gate. Upon reflection, the relative phase between the linear components of the circularly polarized light is conserved but the direction of propagation is reversed, which is equivalent to a reversal of handedness. This reversal of handedness ensures that reflected light that emerges from the quarter-wave plate has a linear polarization that is normal to the incident, linearly polarized light. Therefore, the polarizing beam splitter is used to separate, as much as possible, incident and reflected light. We use a long working distance objective (Mitutoyo M Plan Apo 100X) with a numerical aperture NA = 0.7 to focus and collect light. We use an avalanche photodetector APD430A2 by Thorlabs with a cutoff frequency of 400 MHz.

### Detailed measurement circuits

This section details the circuits used for cross-power measurements of driven vibrations (Fig. 7), vibrational ringdown measurements (Fig. 8) and measurements of phase modulation induced by strain modulation (Fig. 9).

### In-phase and quadrature components of vibrations

The vibrational phase $\phi_\mathrm{m}$ estimated from the in-phase and quadrature components of the photodetector output signal exhibits a linear dependence on drive frequency $f_\mathrm{d}$ away from resonance. This behavior is attributed to propagation delay through the cables. We correct $\phi_\mathrm{m}(f_\mathrm{d})$ by subtracting off this linear dependence. Figure 10a displays $\phi_\mathrm{m}(f_\mathrm{d})$ obtained with approach (i) based on analog demodulation with SDRlab using two mixers and two local oscillators in quadrature. Figure 10b displays $\phi_\mathrm{m}(f_\mathrm{d})$ obtained with approach (ii) based on digital demodulation with SDRlab using one mixer and one local oscillator and $IQ$ demodulation in software. Figure 10c displays $\phi_\mathrm{m}(f_\mathrm{d})$ obtained with approach (iii) based on digital demodulation with an oscilloscope and $IQ$ demodulation in software. Top (bottom) row shows uncorrected (corrected) $\phi_\mathrm{m}$.

### Data availability

The data that support the findings of this study are available from the corresponding author upon reasonable request.

### Code availability

The codes used in this work are available from the corresponding author upon request. The two main codes are shown in Supplementary Note 4.

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

## Acknowledgements

This work was supported by the National Natural Science Foundation of China (grant numbers 62150710547 and 62074107) and the project of the Priority Academic Program Development (PAPD) of Jiangsu Higher Education Institutions. The authors are grateful to Prof. Wang Chinhua for his strong support. The authors wish to thank Prof. R. Picone and the two anonymous referees for the time and the care they took in reviewing the manuscript.

## Author contributions

C.Z. and J.M. assembled the data acquisition system. C.Z., Y.Z. and C.Y. fabricated the device with the help of F.C. H.L., Y.Y. and J.M. built the optical setup. C.Z. and J.M. analyzed the data. J.M. wrote the paper and supervised the work.

## Competing interests

The authors declare no competing interests.
