## [Peer Review File · Communications Engineering]

Reviewers' comments:

Reviewer #1 (Remarks to the Author):

The manuscript discusses several uses of commercial off-the-shelf software defined radio devices in the characterization and control of a nanomechanical resonator.

The manuscript is well written, and in my opinion, is best suited to an audience desiring a thorough case study of a laboratory replacing several dedicated instruments with an inexpensive and flexible software defined radio platform. At least four demonstrations are described: two characterizations of a resonator, controlled excitation of the resonator, and encoding/decoding of a digital transmission through the resonator.

No advancements in the underlying nanomechanical technology are clearly claimed by the authors. The techniques described are broadly applicable, but enough background information about nanomechanical resonators is given for the audience to follow the case study in detail.

The author's own article (reference 55) contains similar work, but this manuscript describes a different apparatus and results.

The encoded/decoded video demonstration is, perhaps, a little gimmicky. The authors omit any discussion of forward error correction and other robust features of the encoding that increase the tolerance of transmission flaws. Figure 6a might suggest zero bit errors; can some quantitative metric of the observed signal quality be given?

Reviewer #2 (Remarks to the Author):

The authors describe in the paper: "A versatile, phase coherent data acquisition system based on software-defined radio for graphene nanomechanics"

the characterization of a graphene nanomechanical drum type Fabry-Perot interferometer formed between the gate electrode and the graphene membrane. The membrane resonant frequency was found to be 77.x MHz.

The abstract promises "Software-defined radios (SDRs) are radio frequency transceivers in which an important part of signal processing is performed digitally using vast libraries of open-source software. Here, we assemble a simple data acquisition system whose architecture, based on SDR, allows us to design a comprehensive suite of tools to study the vibrations of a few-layer graphene nano-mechanical resonator. " Yet multiple electrical and SDR setups are used in the paper like a RED Pitaya, a Nooelec SDR dongle and a Hack-RF One. The papers main focus does not lie in the characterization of the SDR as cheap replacement of high frequency lock-in amplifier or vector network analyzer, it focuses on the characterization of the graphene device. The demonstration of the optical TV-broadcast can be seen as a "gimmick".

A more descriptive title for the paper would be "Optical characterization of a electrostatic driven graphene nanomechanical Fabry-perot interferometer using software-defined radios".

The optical part gets more clear reading then Methods section: "The gap size between the resonator and the top of

the gate electrode is nominally 230 nm to optimize the optical responsivity of the resonator at wavelength $\lambda = 633$ nm. The optical responsivity reads $(\partial PR/\partial z)/P_{in} \approx 3 \times 10^{-3} \text{ nm}^{-1}$, where P_{in} is the optical power incident on the resonator and PR is the optical power re-lected by the device formed by the resonator and the gate."

Why is 230 nm gap size optimizing the optical responsivity? This is quite dependent on the finesse of the Fabry-Perot Resonator! For low finesse this is correct but the authors should add the optical responsivity calculation in the paper, because it is the intensity modulation method. See e.g. attached plot (calculated for a transmissive Fabry- Perot.)

Comments to Section II

Figure 1 should be described in section II. The description of the optical setup is missing completely. The use of the polarizing optics (wave-plates) is necessary and good, but is not described in the paper an not described in reference 1 which is cited for the optical measurement (and even not in the supplementary material of this reference....) : "Its vibrations are driven electrostatically (see Methods) and are detected optically using a well-known technique (Fig. 1a) [1]."

The sentence "The resonator is placed in an optical standing wave formed between a laser source and the gate." is wrong.

Were is the "optical standing wave" formed? We have indeed two resonators: 1. An high finesse electro-mechanical resonator formed by the graphene membrane. 2. A low finesse optical resonator between the graphene membrane and the "gate" electrode. And circular polarized light used to interrogate this resonator.

This should be formulated more precisely.

The caption of figure 1 states "Its in-phase I and quadrature Q components can be extracted in software. Alternatively, I and Q can be measured using analog heterodyning by splitting the signal at v1 (circuit branch in blue)." The difference between the two methods is the possible bandwidth of the measurement. This should be discussed in the paper.

The sentence " While the clock of the receiver cannot be shared, we correct the small frequency offset between the receiver and the external sources in software, a simple process we briefly explain later." is not true. The authors did not implement this. There would be several ways of phase coherent implementations using the hardware chosen by the authors:

1. Using one rf-output of the Red-Pitaya to generate a 10 MHz reference clock.
2. Using a 10 MHz referenced PLL to generate an external 122.88 MHz ADC clock

https://redpitaya.readthedocs.io/en/latest/developerGuide/hardware/122-16_EXT/top.html

So there are multiple options for 10 MHz referenced clocking for all function generators in the setup. So frequency drifts and offsets could be avoided. Even the FPGA could generate clocking using its PLLs
<https://rubiola.org/pdf-articles/journal/2019-UFFC--Red-Pitaya.pdf>

Section II should be clarified and rewritten to clearly describe the optical and electrical setup.

Comments to Section III:

a) The Qm of measurement Fig 2 a) should be fitted and compared with the ring down measurement since they have similar drive power. The errors due to drift should not only be noted but quantified for the actual setup. Usually in averaging processes drifts could be identified using modified Allan-variances.

b)
Ok.

c)
The authors write in this section "We find that (ii) provides the best estimators for I and Q.". Why? The measurement systems should be compared using an error analysis. Why do the signals have a different amount of noise? Usually estimator performance could be estimated using Cramer-Rao bounds if the noise of the signal is analyzed (https://en.wikipedia.org/wiki/Cram%C3%A9r%E2%80%93Rao_bound).

How does a phase error in the IQ mixer setup influence the result (due to cable length mismatch or other effects)?

To be more comparable to a lock-in amplifier IQ and r/ϕ should be plotted ($r=\sqrt{i^2+q^2}$).

d)
Ok.

e)
This section is more "gimmick" than characterization of the device. It shows the possibility of QPSK subcarrier modulation. The constellation diagram should be discussed since it describes the signal quality. It should be discussed in dependence e.g. of the laser power.
The video provided in the supplementary material of the corresponding authors son does not provide any information relevant to the transmission quality (the constellation diagram or an error rate would) - therefore it should not be published to respect the rights of his son.

I would recommend to the authors to optimize the synchronization of the used generators to minimize the drift effects and to discuss the performance limits of the measurement setups. If they do not decide to change the focus of the paper from the SDR to their nano-mechanical device.

Therefore I recommend major revisions before publication.

Reviewer #3 (Remarks to the Author):

General comments:

Overall, this is a good paper and I recommend publication. The authors have built a novel system to measure and control the vibration of an exotic nanomechanical graphene resonator. While there are other methods for making these measurements, their system has some advantages, especially adaptability and low cost. Their presentation of the experimental data demonstrating its effectiveness is good. The encoding and decoding of a video is perhaps superfluous and potentially distracting, but it does serve as a demonstration of the use of the resonator as an information channel. It does have a certain playful quality that readers may find interesting.

Specific comments:

There are missing articles (“a,” “an,” “the”) throughout the text. For instance, “... we measure the cross-spectrum of vibrations in frequency domain ...” should be “... we measure the cross-spectrum of vibrations in the frequency domain”

“To address these fundamental and applied topics, researchers have overcome the challenge of detecting vanishing small vibrations by devising a variety of measurement techniques.”

Perhaps you mean vanishingly small vibrations.

My best to you, the authors, on your future work.

Response to Reviewers (Manuscript COMMSENG-23-0303).

Below, comments from Reviewers appear in blue and the changes we have made in response to the Reviewers' comments appear in red. We also provide a copy of the new main text where these changes are highlighted in red.

Reviewer #1 (Remarks to the Author):

We are grateful to the Reviewer for carefully reviewing our manuscript.

The Reviewer writes :

The manuscript discusses several uses of commercial off-the-shelf software defined radio devices in the characterization and control of a nanomechanical resonator.

The manuscript is well written, and in my opinion, is best suited to an audience desiring a thorough case study of a laboratory replacing several dedicated instruments with an inexpensive and flexible software defined radio platform. At least four demonstrations are described: two characterizations of a resonator, controlled excitation of the resonator, and encoding/decoding of a digital transmission through the resonator.

We thank the Reviewer for the positive comments and for the Reviewer's general appreciation of our manuscript.

The Reviewer writes :

No advancements in the underlying nanomechanical technology are clearly claimed by the authors.

We thank the Reviewer for this comment. We believe that our strain-induced phase modulation study is a novelty. We consulted several experts in the field of nanomechanics about it, who told us that they were not aware of existing works showing a decrease of the amplitude of phase modulation as strain modulation frequency increases. To emphasize this point, on page 15 of the new version of the main text, Section "D. Phase modulation induced by strain modulation", we now write:

Here, we measure ϕ_m in the time domain in response to a modulated strain with various modulation frequencies and modulation strengths. To the best of our knowledge, such a study has not previously been carried out.

The Reviewer writes :

The author's own article (reference 55) contains similar work, but this manuscript describes a different apparatus and results.

We thank the Reviewer for mentioning our previous work. On page 4 of the new version of the main text, we now write:

Implementing this important feature [phase coherence of the signal acquisition] was not possible in our earlier work [27], where we employed a single-channel, narrow-band SDR dongle as a radio frequency power meter to measure large amplitude vibrations in a different type of FLG resonator.

The Reviewer writes :

The encoded/decoded video demonstration is, perhaps, a little gimmicky. The authors omit any discussion of forward error correction and other robust features of the encoding that increase the tolerance of transmission flaws. Figure 6a might suggest zero bit errors; can some quantitative metric of the observed signal quality be given?

We thank the Reviewer for commenting on our video demonstration and for suggesting that we quantify the robustness of our encoding. Motivated by the Reviewer's suggestions, we are now revisiting our video encoding work to study its tolerance to transmission flaws. We hope that our new data will lead to a self-contained study that may warrant a future, separate publication. For this reason, we would like to remove the video demonstration and the related discussion from the current manuscript.

Once again, we thank the Reviewer for taking the time to carefully read our manuscript and for offering helpful comments.

Reviewer #2 (Remarks to the Author):

We are grateful to the Reviewer for carefully reading our manuscript and for offering many valuable comments and suggestions. Addressing the Reviewer's comments enabled us to significantly improve our manuscript.

The Reviewer writes:

The authors describe in the paper: "A versatile, phase coherent data acquisition system based on software-defined radio for graphene nanomechanics" the characterization of a graphene nanomechanical drum type Fabry-Perot interferometer formed between the gate electrode and the graphene membrane. The membrane resonant frequency was found to be 77.x MHz.

The abstract promises "Software-defined radios (SDRs) are radio frequency transceivers in which an important part of signal processing is performed digitally using vast libraries of open-source software. Here, we assemble a simple data acquisition system whose architecture, based on SDR, allows us to design a comprehensive suite of tools to study the vibrations of a few-layer graphene nano-mechanical resonator. " Yet multiple electrical and SDR setups are used

in the paper like a RED Pitaya, a Nooelec SDR dongle and a Hack-RF One.

We thank the Reviewer for this comment. As explained below, we have decided to shorten our manuscript and to remove the last part about our video demonstration. As a result, the new version of our manuscript no longer mentions Nooelec SDR dongle or HackRF One.

The Reviewer writes :

The papers main focus does not lie in the characterization of the SDR as cheap replacement of high frequency lock-in amplifier or vector network analyzer, it focuses on the characterization of the graphene device. The demonstration of the optical TV-broadcast can be seen as a "gimmick".

A more descriptive title for the paper would be "Optical characterization of a electrostatic driven graphene nanomechanical Fabry-perot interferometer using software-defined radios".

We thank the Reviewer for this comment. We agree that the focus of our work is the thorough characterization of a graphene nanomechanical resonator using a simple, phase-coherent software-defined radio. We like the title proposed by the Reviewer, as it neatly captures the essence of our work. We propose to simplify it slightly, as follows:

Graphene nanomechanical vibrations measured with a phase-coherent software-defined radio.

The Reviewer writes :

The optical part gets more clear reading then Methods section: "The gap size between the resonator and the top of the gate electrode is nominally 230 nm to optimize the optical responsivity of the resonator at wavelength $\lambda = 633$ nm. The optical responsivity reads $(\partial PR/\partial z)/P_{in} \approx 3 \times 10^{-3} \text{ nm}^{-1}$, where P_{in} is the optical power incident on the resonator and PR is the optical power reflected by the device formed by the resonator and the gate."

Why is 230 nm gap size optimizing the optical responsivity? This is quite dependent on the finesse of the Fabry-Perot Resonator! For low finesse this is correct but the authors should add the optical responsivity calculation in the paper, because it is the intensity modulation method. See e.g. attached plot (calculated for a transmissive Fabry- Perot.)

We thank the Reviewer for suggesting that we explain how the optical responsivity is calculated. We also thank the Reviewer for sending the interesting transmittance response of the Fabry-Perot interferometer as a function of wavelength where two different finesses are used. To address the Reviewer's comment, **we created a new section in Supplementary Information entitled "IV. Reflectance and absorbance of FLG suspended over a reflective substrate."**

To summarize this new section, in it we explain how we model our device as a multilayered structure across which normally incident, monochromatic, electromagnetic plane waves

propagate. We employ a transfer matrix technique to calculate the complex amplitudes of the forward and backward moving electric field waves across the structure. We calculate the reflectance of the device, the derivative of which with respect to resonator displacement shows a maximum where the cavity depth (that is, the distance between few-layer graphene FLG and the gate electrode) is 217 nm. Because FLG membranes tend to sag ~ 10 nm inside cavities after being transferred onto their substrate, we design the cavity depth to be a little larger and choose it to be 230 nm.

We created a new Fig. S3 to depict the multilayered structure and the optical standing wave.

FIG. S3. Multilayered structure and optical standing wave. 0: vacuum; 1: FLG; 2: vacuum; 3: Au; 4: SiO₂; 5: Si. $E_{q\pm}$ ($E'_{q\pm}$) are the complex amplitudes of forward + and backward - propagating electric field waves on the left side (on the right side) of the interface between layer $q - 1$ and layer q . Arrows indicate the directions of propagation (the field vectors are perpendicular to the directions of propagation and parallel to the interfaces). The plot shows the magnitude of the total electric field $|E(z)| = |E_+(z) + E_-(z)|$ normalized to the magnitude of the incident electric field E_{1+} . Thicker grey trace: no FLG. Thinner blue trace: FLG placed at $|z_{eq}| = 217$ nm from the surface of the substrate at $z = 0$.

We also created a new Fig. S4 to show the reflectance, the absorbance, and the gradient of reflectance of FLG as a function of the distance z_{eq} between FLG and the substrate (please see next page).

FIG. S4. Reflectance R (red), absorbance A (blue) and gradient of reflectance $|dR/dz|$ of FLG as a function of distance between FLG and the substrate z_{eq} . Here, FLG is made of 3 layers of graphene. $R + A < 1$ because the substrate is not fully reflective.

The Reviewer writes :

Figure 1 should be described in section II. The description of the optical setup is missing completely. The use of the polarizing optics (wave-plates) is necessary and good, but is not described in the paper and not described in reference 1 which is cited for the optical measurement (and even not in the supplementary material of this reference...): "Its vibrations are driven electrostatically (see Methods) and are detected optically using a well-known technique (Fig. 1a) [1]."

We thank the Reviewer for suggesting that we describe the optical setup. We were surprised by the fact that a full description of the optical setup is indeed difficult to find in the literature. To correct this, **we complemented our ‘Methods’ section** with the following text:

Optical detection of vibrations. We use a Helium-Neon laser as an optical source emitting at $\lambda = 633$ nm. We filter the output of the laser with a single mode fiber and control its linear polarization with a half-wave plate. We use a combination of a polarizing beam splitter and a quarter-wave plate to steer the beam reflected off the device into the photodetector, as follows. Linearly polarized light incident on the surface of the quarter-wave plate facing the polarizing beam splitter (Fig. 1a) becomes circularly polarized upon traversing the quarter-wave plate. It then propagates to the device, where it gets reflected. Upon reflection, the relative phase between the linear components of the circularly polarized light is conserved but the direction of propagation is reversed, which is equivalent to a reversal of handedness. This reversal of handedness ensures that reflected light that emerges from the quarter-wave plate has a linear polarization that is normal to the incident, linearly polarized light. Therefore, the polarizing beam splitter is used to separate, as much as possible, incident and reflected light. We use a long working distance objective (Mitutoyo M Plan Apo 100X) with a numerical aperture

$NA = 0.7$ to focus and collect light. We use an avalanche photodetector APD430A by Thorlabs with a cutoff frequency of 400 MHz.

We also moved the description of Fig. 1 to Section II.

The Reviewer writes :

The sentence "The resonator is placed in an optical standing wave formed between a laser source and the gate." is wrong. Where is the "optical standing wave" formed? We have indeed two resonators: 1. A high finesse electro-mechanical resonator formed by the graphene membrane. 2. A low finesse optical resonator between the graphene membrane and the "gate" electrode. And circular polarized light used to interrogate this resonator. This should be formulated more precisely.

We thank the Reviewer for suggesting that we explain where the optical standing wave forms in our setup. This suggestion was an eye opener. Surely a standing wave exists as a superposition of forward and backward moving electric field waves, but as the Reviewer rightly points out, that standing wave does not establish itself between the laser and the gate. To address the Reviewer's comment, **in Supplementary Information we added a discussion about the optical standing wave to our new section IV. Reflectance and absorbance of FLG suspended over a reflective substrate.** In it, we write:

To optimize the transduction of FLG vibrations into a modulated optical signal, we model our device as a multilayered structure (Fig. S3, top panel) across which normally incident, monochromatic, electromagnetic plane waves propagate. We consider the simplest case of a linear polarization for the real-valued electric field $\text{Re}[\mathbf{E}(z, t)]$, where z is the position along the direction perpendicular to the structure and t is time. Even though incident and reflected waves are circularly polarized between the device and the surface of the quarter-wave plate facing the device, these waves do not traverse any birefringent medium and travel along the same optical path. As a result, incident and reflected waves superimpose themselves on one another and interfere as linearly polarized waves would. The reason for this is that (i) circularly polarized plane waves can be decomposed into two orthogonal, linearly polarized waves; (ii) each of the two incident, linearly polarized waves superimposes on its reflected, linearly polarized wave. We employ a transfer matrix technique to calculate the complex amplitudes E_+ , E_- of the forward and backward moving electric field waves (that is, the complex amplitudes of the normal modes of the electromagnetic field) across the multilayered structure [4]. (...) By propagating $E_{\pm}(z)$ through the structure and matching them across interfaces, we calculate the total electric field $E(z) = E_+(z) + E_-(z)$ as the superposition of forward and backward moving electric field waves. In a steady state, this superposition of propagating fields leads to the buildup of a standing wave, which is located between the device and the surface of the quarter-wave plate facing the device (elsewhere, the linear polarizations of incident and reflected light are orthogonal). The magnitude of this standing wave $|E(z)|$ is shown in Fig. S3, lower panel, both in the case where FLG is absent and in the case where FLG is placed $z_{\text{eq}} = 217$ nm away from the surface of the substrate (see below for the estimation of z_{eq}).

FLG mostly acts as an optical absorber (see below) and perturbs the shape of the standing wave only weakly.

We also modified the main text. On page 5, we now write:

We briefly describe our resonator and our vibration detection setup below. FLG is made of 3 to 4 graphene layers (Supplementary Information, Section III). The resonator is shaped as a drum and is suspended over a local gate electrode, Figs. 1b-d (see Methods). It is placed in vacuum and is measured at room temperature. Its vibrations are driven electrostatically (see Methods) and are detected optically (see Fig. 1a, Methods, and Ref. [1]). The resonator is placed in an optical standing wave formed between the nanomechanical device and the surface of a quarter-wave plate facing the device (Supplementary Information, Section IV).

The Reviewer writes :

The caption of figure 1 states "Its in-phase I and quadrature Q components can be extracted in software. Alternatively, I and Q can be measured using analog heterodyning by splitting the signal at v_I (circuit branch in blue)." The difference between the two methods is the possible bandwidth of the measurement. This should be discussed in the paper.

We thank the Reviewer for commenting on the difference between the two approaches used to demodulate I and Q . On page 14 of the new version of the manuscript, we now write:

We note that our technique of demodulating I and Q in software and the previous technique based on using two mixers have a similar measurement bandwidth. This is because the software technique still requires a mixer for down-conversion, and because in both cases most of the measurement time is spent downloading data to the host computer and averaging them. Overall, measuring averaged I and Q spectra with either technique requires the same amount of time.

The Reviewer writes :

The sentence " While the clock of the receiver cannot be shared, we correct the small frequency offset between the receiver and the external sources in software, a simple process we briefly explain later." is not true. The authors did not implement this. There would be several ways of phase coherent implementations using the hardware chosen by the authors:

1. Using one rf-output of the Red-Pitaya to generate a 10 MHz reference clock.
2. Using a 10 MHz referenced PLL to generate an external 122.88 MHz ADC clock https://redpitaya.readthedocs.io/en/latest/developerGuide/hardware/122-16_EXT/top.html

So there are multiple options for 10 MHz referenced clocking for all function generators in the setup. So frequency drifts and offsets could be avoided. Even the FPGA could generate clocking using its PLLs <https://rubiola.org/pdf-articles/journal/2019-UFFC--Red-Pitaya.pdf>

Section II should be clarified and rewritten to clearly describe the optical and electrical setup.

We thank the Reviewer for suggesting that we elaborate on possibilities to improve our setup and synchronize the clock of the receiver with the clocks of other instruments in the setup. On page 5 of the new version of the manuscript, we now write:

The results shown below are obtained in the case where the receiver's analog-to-digital converters are driven by a separate clock. In Supplementary Information, Section II, we present spectra of vibrational amplitude and vibrational phase measured in configurations where the receiver and the other instruments share the same clock. At room temperature, where the linewidths of vibrational spectra are large, possible offsets between clocks do not adversely affect measurements. At cryogenic temperature, however, where vibrational spectra are narrow, clock synchronization is essential.

To address the Reviewer's comments in full, **we purchased a new SDRlab instrument** that must be used with an external clock (SDRlab 122-16 External Clock Standard Kit). With it, we measured new data, and **created a new section in Supplementary Information** entitled “**II. Synchronizing the clocks of instruments.**” This new section reads:

Below, we address the topic of clock synchronization among the receiver, the driving source, and the local oscillator used for down-conversion. To illustrate the discussion, we present measurements of the vibrational amplitude and of the vibrational phase as a function of drive frequency f_d , carried out with 3 different clock configurations. Data were obtained using the technique of demodulating I and Q in software with SDRlab and may be compared with data in Figs. 4g, h in the main text. As in Fig. 4, the same device was measured at room temperature, the drive power was $P_d = -40$ dBm, the gate voltage was $V_g^{\text{dc}} = -15$ V, the incident optical power was $P_{\text{inc}} = 3$ μ W, and measured signals were down-converted to ~ 1 MHz and averaged 500 times. However, data in this section were measured several months after those in Fig. 4, after the chip holding the device had undergone several manipulations. Data in Figs. S1a, b were obtained using a 0 dBm, 10 MHz signal at the output of SDRLab that was used as a clock for all instruments. Data in Figs. S1c, d were obtained using a different type of SDRlab instrument that has to be used with an external clock at 122.8 MHz (SDRlab 122-16 External Clock Standard Kit). To supply the clock signal, we used a low-noise signal generator from Rohde & Schwarz, and connected its 10 MHz frequency reference output port to the 10 MHz frequency reference input ports of the driving source and of the local oscillator. Data in Figs. S1e, f were obtained with our standard SDRlab receiver running on its own clock, as in Fig. 4g, h. Data shown in Figs. S1a-f were obtained within the same measurement session. As in Fig. 4, solid traces are fits to the response of a harmonic oscillator. We find that the 3 different clock configurations yield qualitatively similar results. At room temperature, where spectral linewidths are large, possible offsets between clocks do not adversely affect measurements. At cryogenic temperature, however, where vibrational spectra are narrow, clock synchronization is essential and would have to be implemented.

Please see Fig. S1 on the next page.

Fig. S1. Synchronizing the clocks of instruments. (a), (b) A 0 dBm, 10 MHz signal at the output of SDRLab was used as a clock for all instruments. (c), (d) SDRLab 122-16 External Clock Standard Kit was used with the clock signal supplied by a signal generator from Rohde & Schwarz. (e), (f) The standard SDRLab receiver running on its own clock was used.

The Reviewer writes :

a) The Q_m of measurement Fig 2 a) should be fitted and compared with the ring down measurement since they have similar drive power.

We thank the Reviewer for this comment. In the first version of our manuscript, we already discussed that our spectral measurements and our vibrational ringdown measurements yield the same estimation for Q_m . On page 9, the main text reads:

Having a good estimator for the driven response is important to measure the quality factor Q_m of the vibrational mode. The latter can be estimated from the response in frequency, $Q_m = f_m/\Delta f$, where f_m is the resonant frequency of the mode and Δf is the full width at half maximum of the power response. However, nondissipative spectral broadening due to resonant frequency fluctuations [35–44] may result in an underestimate of Q_m . Ringdown experiments, where the drive is suddenly switched off and vibrations are left to freely decay, offer an unambiguous estimation of Q_m as they solely measure energy [45–49]. By integrating a radio frequency switch in our drive circuit (Supplementary Information, Section V), we performed such ringdown experiments. As shown in Fig. 2f, the amplitude of vibrations down-converted to 8.2 MHz and averaged 500 times decays as $\sim \exp[-t/(2\tau)]$ with $\tau = Q_m/(2\pi f_m)$. Using $f_m = 77.6$ MHz we find $Q_m \cong 120$, which coincides with the estimate based on the resonance linewidth (Fig. 2a). This result agrees with earlier ringdown measurements in MoS₂ resonators, which showed that at room temperature the linewidth of the resonance is mostly accounted for by energy dissipation, with no visible contribution from frequency fluctuations [46].

In the context of the ringdown experiment, the Reviewer writes :

The errors due to drift should not only be noted but quantified for the actual setup. Usually in averaging processes drifts could be identified using modified Allan-variances.

We thank the Reviewer for this comment. On page 9 of the new version of the manuscript, we now elaborate on our ringdown experiments and write:

We also verified that the averaging process does not affect our estimation of τ . Indeed, phase and frequency noise in the signal would lower the amplitude of the averaged driven signal but would not modify the averaged exponential decay. Further, we measured the time jitter between subsequent trigger events and found it to be at most 10 ns; our numerical calculations show that this time jitter does not affect the averaged exponential decay since $\tau \cong 250$ ns.

Furthermore, because of the broad linewidth of the vibrational spectral response, it was not possible for us to detect drift in the resonant frequency of vibrations, which we could have analyzed using the Allan variance.

The Reviewer writes :

c) The authors write in this section "We find that (ii) provides the best estimators for I and Q ". Why?

We thank the Reviewer for this comment. On page 12 of the new version of the main text, we have removed the sentence "We find that (ii) provides the best estimators for I and Q ." Instead, we now write:

We find that (ii) yields I and Q spectra that most resemble the frequency response of a harmonic oscillator.

To elaborate on this statement, on page 14 of the main text we discuss configuration (i), which is based on 2 mixers and 2 local oscillators, and write:

We suspect that deviations from the [harmonic oscillator] model are caused by slight phase and amplitude imbalances between the two local oscillators (...).

As a follow-up comment, the Reviewer writes :

The measurement systems should be compared using an error analysis. Why do the signals have a different amount of noise? Usually estimator performance could be estimated using Cramer-Rao bounds if the noise of the signal is analyzed (https://en.wikipedia.org/wiki/Cram%C3%A9r%E2%80%93Rao_bound).

On page 15 of the new version of the main text, we now write:

In addition, 4096 time traces are acquired and averaged by the oscilloscope, compared with 500 averages with SDRlab (which may explain why $v_1(t)$ in Fig.~4i displays a lesser amount of noise than $v_1(t)$ in Fig.~4e).

We also thank the Reviewer for mentioning the Cramér–Rao bound. We will consider it in our future work.

The Reviewer writes :

How does a phase error in the IQ mixer setup influence the result (due to cable length mismatch or other effects)?

We thank the Reviewer for suggesting that we discuss the impact of phase error in the IQ mixer setup on the measurements of I and Q . On page 12 of the new version of the main text, we now write:

Note that $v_A^2 + v_B^2 = (I^2 + Q^2)\alpha^2/8$, which means that the measurement of the vibrational

energy does not depend on the demodulation procedure (within an attenuation factor) if ϕ' is the same for both v_A and v_B and if the two local oscillators are at the same frequency.

In addition, on page 14 of the new version of the main text, we now elaborate on configuration (i) based on 2 mixers and 2 local oscillators and we write:

We suspect that deviations from the model are caused by slight phase and amplitude imbalances between the two local oscillators, which result in $v_A^2 + v_B^2 \neq (I^2 + Q^2)\alpha^2/8$ and in distorted $\tilde{I}(f_d)$, $\tilde{Q}(f_d)$.

The Reviewer writes :

To be more comparable to a lock-in amplifier IQ and r/phi should be plotted ($r=\sqrt{i^2+q^2}$).

We thank the Reviewer for this comment. We prefer the format of Fig. 4, where the spectra of I and Q are shown. As far as we know, it is a common format in the nanomechanics community (see, e.g., Fig. 2b, c in Ref. [50]).

The Reviewer writes :

e) This section is more "gimmick" than characterization of the device. It shows the possibility of QPSK subcarrier modulation. The constellation diagram should be discussed since it describes the signal quality. It should be discussed in dependence e.g. of the laser power. The video provided in the supplementary material of the corresponding authors son does not provide any information relevant to the transmission quality (the constellation diagram or an error rate would) - therefore it should not be published to respect the rights of his son.

We thank the Reviewer for commenting on our video demonstration. We are now revisiting our video encoding work to study its tolerance to transmission flaws. We hope that our new data will lead to a self-contained study that may warrant a future, separate publication. For this reason, we would like to remove the video demonstration and the related discussion from the current manuscript.

The Reviewer writes :

I would recommend to the authors to optimize the synchronization of the used generators to minimize the drift effects and to discuss the performance limits of the measurement setups. If they do not decide to change the focus of the paper from the SDR to their nano-mechanical device.

Therefore I recommend major revisions before publication.

We hope that we have now aptly addressed all the Reviewer's comments. Once again, we thank the Reviewer for taking the time to carefully read our manuscript and for offering many helpful comments and suggestions.

Reviewer #3 (Remarks to the Author):

We thank Prof. Picone for taking the time to carefully review our manuscript.

Prof. Picone writes:

General comments:

Overall, this is a good paper and I recommend publication. The authors have built a novel system to measure and control the vibration of an exotic nanomechanical graphene resonator. While there are other methods for making these measurements, their system has some advantages, especially adaptability and low cost. Their presentation of the experimental data demonstrating its effectiveness is good. The encoding and decoding of a video is perhaps superfluous and potentially distracting, but it does serve as a demonstration of the use of the resonator as an information channel. It does have a certain playful quality that readers may find interesting.

We are grateful to Prof. Picone for his positive and encouraging comments. Given that it may be superfluous and distracting, we would like to remove the video demonstration (and the related discussion) from the current manuscript. We are now revisiting our video encoding work to study its tolerance to transmission flaws. We hope that our new data will lead to a self-contained study that may warrant a future, separate publication.

Prof. Picone writes:

Specific comments:

There are missing articles (“a,” “an,” “the”) throughout the text. For instance, “... we measure the cross-spectrum of vibrations in frequency domain ...” should be “... we measure the cross-spectrum of vibrations in the frequency domain”

*“To address these fundamental and applied topics, researchers have overcome the challenge of detecting vanishing small vibrations by devising a variety of measurement techniques.”
Perhaps you mean vanishingly small vibrations.*

We thank Prof. Picone for this comment. We have scrutinized our manuscript for missing articles and have corrected typos.

Prof. Picone writes:

My best to you, the authors, on your future work.

We thank Prof. Picone very much for reading our manuscript carefully and for offering encouraging comments.

REVIEWERS' COMMENTS:

Reviewer #2 (Remarks to the Author):

The authors did great improvements to their paper. It is much more focused and the technical description of the experiments can be used to reproduce the results. Therefore I recommend the publication. Great job.

Reviewer #3 (Remarks to the Author):

I find the authors' response to me adequate and recommend publication. My congratulations on an excellent work.